# Engineering a pure Dirac regime in ZrTe$_5$

Jorge I. Facio[1,2,3], Elisabetta Nocerino[4], Ion C. Fulga[2], Rafal Wawrzynczak[5],
Joanna Brown[5], Genda Gu[6], Qiang Li[6,7], Martin Mansson[4], Yasmine Sassa[8],
Oleh Ivashko[9], Martin v. Zimmermann[9], Felix Mende[10], Johannes Gooth[5,11],
Stanislaw Galeski[5,11], Jeroen van den Brink[2,10], and Tobias Meng[10]⋆

**1** Centro Atómico Bariloche and Instituto Balseiro, CNEA, 8400 Bariloche, Argentina
**2** Institute for Theoretical Solid State Physics, IFW Dresden and Würzburg-Dresden Cluster of Excellence ct.qmat, Helmholtzstr. 20, 01069 Dresden, Germany
**3** Instituto de Nanociencia y Nanotecnología CNEA-CONICET, Argentina
**4** Department of Applied Physics, KTH Royal Institute of Technology,
SE-100 44, Stockholm, Sweden
**5** Max Planck Institute for Chemical Physics of Solids, 01187 Dresden, Germany
**6** Condensed Matter Physics and Materials Science Division, Brookhaven National
Laboratory, Upton, New York 11973-5000, USA,
**7** Department of Physics and Astronomy, Stony Brook University, Stony Brook,
NY 11794-3800, USA,
**8** Department of Physics, Chalmers University of Technology, SE-412 96 Göteborg, Sweden
**9** Deutsches Elektronen-Synchrotron DESY, Notkestr. 85, 22607 Hamburg, Germany
**10** Institute of Theoretical Physics and Würzburg-Dresden Cluster of Excellence ct.qmat,
Technische Universität Dresden, 01069 Dresden, Germany
**11** Physikalisches Institut, Universität Bonn, Nussallee 12, D-53115 Bonn, Germany

⋆ tobias.meng@tu-dresden.de

## Abstract

Real-world topological semimetals typically exhibit Dirac and Weyl nodes that coexist with trivial Fermi pockets. This tends to mask the physics of the relativistic quasiparticles. Using the example of ZrTe$_5$, we show that strain provides a powerful tool for in-situ tuning of the band structure such that all trivial pockets are pushed far away from the Fermi energy, but only for a certain range of Van der Waals gaps. Our results naturally reconcile contradicting reports on the presence or absence of additional pockets in ZrTe$_5$, and provide a clear map of where to find a pure three-dimensional Dirac semimetallic phase in the structural parameter space of the material.

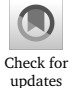

# 1  Introduction

Dirac cones are formed by states that have a linear dispersion relation close to a band touching point. These electronic states have large velocities, long wavelengths, and are robust against backscattering. Materials displaying pure Dirac physics, meaning that there are no additional pockets at the Fermi level, are thus promising candidates for various technological applications.

The age of Dirac semimetals started with graphene, which provides a concrete, pure Dirac system in two dimensions (2D). Even though the bands of graphene are actually gapped due to spin-orbit coupling, the smallness of this gap when compared to temperature and the linear nature of the bands renders graphene physics "Dirac-like."

While graphene is well-known and experimentally available, finding materials that show pure Dirac physics in in three dimensions (3D) has proven to be a challenge. Various mechanisms to accomplish a Dirac cone in 3D have been identified, all of them bringing associated challenges [1]. One natural approach is to stack weakly-coupled 2D systems known to yield 2D Dirac cones [2]. This approach has recently been found successful, e.g., in Kagome metals [3,4], which, however, often present massive Dirac cones not necessarily close to the Fermi energy. A second approach relies on looking for symmetry-enforced crossings between four bands. While this mechanism is arguably the most desirable, as it yields stable Dirac cones, it is often the case that other trivial pockets are also present at relevant energies [2]. A different route is based on the band inversion mechanism. This is predicted to be the root cause in the perhaps most established 3D Dirac semimetal families, $A_3B$ where A =(Na, K, Rb) and B=(As, Sb, Bi) [5,6]; and $Cd_3As_2$ [7–10]. In these cases, while the Dirac cones are not enforced by symmetry, the crystal symmetry still plays a role by avoiding the emergence of a mass term that would yield an insulating phase. Last, also rooted in the band inversion, Dirac cones can be stabilized at $Z_2$ topological phase transitions in centrosymmetric compounds [11,12]. This class does require fine tuning to reach a precisely gapless Dirac semimetal, but it is of interest since there exist materials which are naturally close to such a transition. Then, similar to the case of the slightly gapped graphene, there would exist a range in parameter space around the gap closing point where interesting Dirac physics is within reach of experiments, for instance because the chemical potential is in the linear band regime.

The latter is believed to be the case of $ZrTe_5$, a 3D material with a structural lattice layered along the crystallographic $b$- and $c$-directions. Its electronic structure is close to a transition from a weak topological insulator (WTI) to a strong topological insulator (STI) phase occurring

via the formation of a bulk Dirac node [13–27]. The low-energy band structure of $ZrTe_5$ can therefore be approximated by a (weakly) gapped Dirac Hamiltonian [25]. Experimental evidence suggests that most samples might be in the WTI regime [28–31, 31–34], but that the sign of the Dirac gap is in general sample-dependent [35]. Additionally, the chemical potential is typically larger than the bulk gap, and $ZrTe_5$ samples are therefore often metals with low carrier densities.

The position of the chemical potential shifts with temperature [36], and a Lifshitz transition between electron and hole bands has been reported [24, 30, 37–39]. Furthermore, different quantum oscillation measurements as well as different *ab-initio* studies lead to conflicting conclusions on the presence or absence of additional pockets [30, 40–42]. Metallic $ZrTe_5$ samples typically show rich (magneto-) transport properties, including a large negative magnetoresistance [16], exotic thermoelectric responses [39, 43–45], a large magnetochiral anisotropy [46], a nontrivial field dependence of ultrasound [47], and an unusual Hall response [21, 48, 49] that may or may not indicate the formation of a charge density wave [24, 38, 50–54]. In addition, measurements of $ZrTe_5$ and its sister compound, $HfTe_5$, suggest that electron-electron interactions may favor the formation of more exotic states at large magnetic fields [38, 55, 56], and that external pressure can drive superconducting instabilities [57].

Recent work showed that strain [22, 58, 59], and even phonons [60], can drive $ZrTe_5$ from an STI to a WTI phase (coherent infrared phonons can also induce a Weyl semimetal phase [61]). However, as we discuss below, in the case of the recently studied uniaxial strain along the $a$-direction [22, 58, 59], this transition can be masked by additional, trivial, pockets that lower their energy as strain is applied, yielding an overall semimetallic phase at large strain. Our calculations indicate that whether this scenario or the more conventional insulator-to-insulator transition takes place is determined by a crystal structure parameter largely unaffected by strain: the Van der Waals gap of the crystal along the $c$-direction. Combining synchrotron x-ray diffraction data with *ab-initio* calculations, we show that (i) a pure Dirac system can be achieved by uniaxial strain, but that (ii) the experimental observability of pure Dirac physics strongly depends on the size of the Van der Waals gaps in the crystal structure.

The remainder of the manuscript is organized as follows. Section 2 reports on synchrotron diffraction data that allows us to identify the microscopic atomic positions within the unit cells of our $ZrTe_5$ samples. In Sec. 3, we present *ab-inito* simulations that suggest the electronic band structure to rather sensitively depend on atomic positions, as well as on applied strain. These numerical findings are given a physical interpretation in Sec. 4, where we identify a specific structural Van der Waals gap in the crystal as the key player in determining the material's band structure. Based on our microscopic understanding of how atomic positions impact the band structure, we predict when a pure Dirac phase occurs in $ZrTe_5$ as a function of strain and of the Van der Waals gap in Sec. 5. Our conclusions are finally summarized in Sec. 6.

## 2  Diffraction data

Our starting point is a detailed analysis of the structural properties of unstrained $ZrTe_5$ crystals. To that end, crystal synchrotron x-ray diffraction measurements were performed at the broad band diffraction beam-line P21.1 at PETRA III in DESY (Germany) on a ZrTe5 crystal previously characterized in Ref. [24]. Three-dimensional maps of the reciprocal space were collected with an angular step width of $\Delta\omega = 0.1°$ at 10 K. The data reduction and unit cell determination were carried out with the software CrysAlis$^{Pro}$ [62]. Figure 1 shows the reciprocal lattice plane (h 0 l) at a temperature of 10 K with the overlapped reciprocal lattice grid obtained from the calculated unit cell $a = 3.94827(11)$ Å, $b = 14.3523(8)$ Å, $c = 13.5610(5)$ Å.

As shown in Fig. 1, this unit cell is able to correctly index the Bragg reflections in the

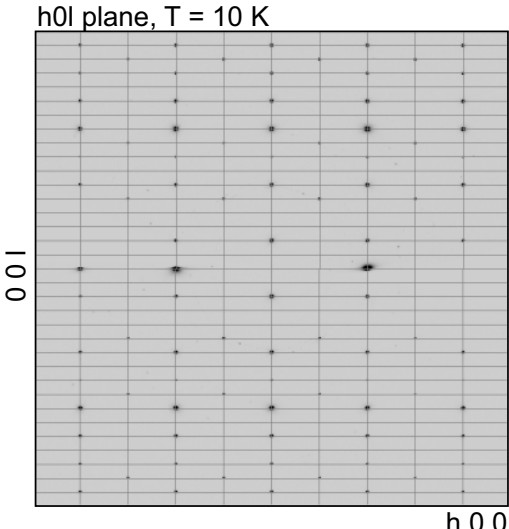

Figure 1: Reconstructed reciprocal space plane (h 0 l) at 10 K. The reciprocal lattice grid extracted from the calculated unit cell provides a correct indexing of the measured Bragg peaks.

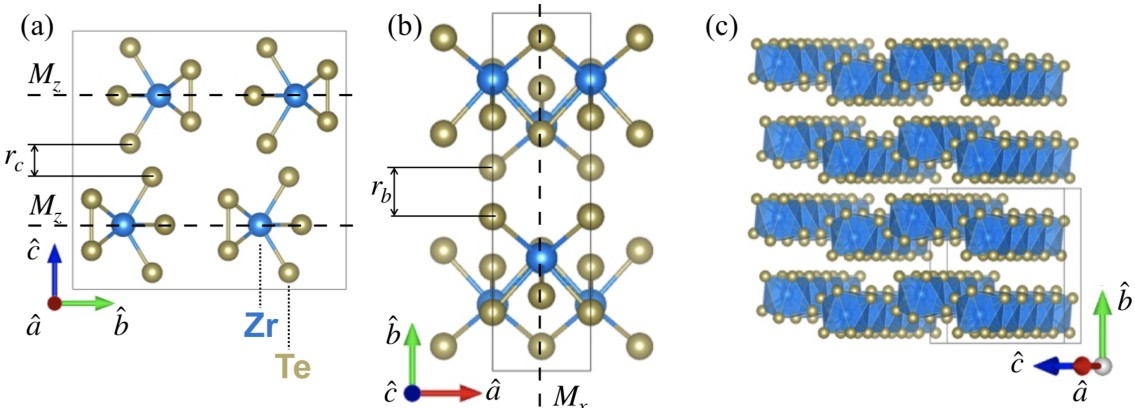

Figure 2: (a,b) Refined crystal structure of $ZrTe_5$: $b-c$ plane and $a-b$ plane views, respectively. (c) Polyhedral representation of the $ZrTe_5$ atomic structure. The 1D chains along the $a$-axis and 2D planes stacked along the $b$-axis are clearly visible. Dashed lines indicate the position of reflection symmetry planes.

dataset. The crystal system was determined to be orthorhombic, with space group $Cmcm$ (63) in standard setting. The atomic positions were refined with the software Olex/Shelx [63]. The Zr atoms fully occupy the $4c$ crystallographic site while the $Te_n$ atoms occupy the $4c$ and $8f$ sites. The resulting low-temperature crystal structure is shown in Fig. 2. Our samples exhibit $ZrTe_8$ edge-sharing polyhedra forming one-dimensional (1D) chains along the $a$-axis, which in turn combine into planes stacked along the $b$-axis. Table 1 presents the lattice parameters and atomic coordinates as obtained from our own refinement at 10 K (structure $A$). The reliability of our refinement is underlined by the low values of the agreement R factors which is equal to 6.85%. In the following, the lattice parameters and atomic coordinates we obtained will be compared to the ones reported in Ref. [64] (structure $B$), which for convenience are repeated in Tab. 1.

Table 1: Crystal structural parameters (Space Group $Cmcm$ (63)). Lattice parameters and Van der Waals gaps are written in units of Å.

| Refinement for structure $A$ (own data at 10 K) | | | | | |
|---|---|---|---|---|---|
| $a$ | $b$ | $c$ | $\alpha$ | $\beta$ | $\gamma$ |
| 3.94827(11) | 14.3523(8) | 13.5610(5) | 90 | 90 | 90 |
| $r_b$ | 1.9505 | | $r_c$ | 1.7426 | |
| Atomic position | $x$ | $y$ | $z$ | Occupancy | Site |
| Zr | 0 | 0.31560(2) | 0.25 | 1 | 4c |
| $Te_1$ | 0 | 0.66324(2) | 0.25 | 1 | 4c |
| $Te_2$ | 0 | 0.93205(2) | 0.14945(2) | 1 | 8f |
| $Te_3$ | 0 | 0.20976(2) | 0.43574(2) | 1 | 8f |
| Refinement for structure $B$ (from Ref. [64]) | | | | | |
| $a$ | $b$ | $c$ | $\alpha$ | $\beta$ | $\gamma$ |
| 3.9875 | 14.53 | 13.724 | 90 | 90 | 90 |
| $r_b$ | 1.9470 | | $r_c$ | 1.8115 | |
| Atomic position | $x$ | $y$ | $z$ | Occupancy | Site |
| Zr | 0 | 0.316 | 0.25 | 1 | 4c |
| $Te_1$ | 0 | 0.663 | 0.25 | 1 | 4c |
| $Te_2$ | 0 | 0.933 | 0.151 | 1 | 8f |
| $Te_3$ | 0 | 0.209 | 0.434 | 1 | 8f |

## 3 Sensitivity of the band structure to atomic positions and strain

To determine how lattice parameters and atomic positions influence the low-energy band structure of $ZrTe_5$, we feed both experimentally determined crystal structures $A$ and $B$ into an *ab-initio* calculation. We perform density-functional theory (DFT) calculations using the FPLO code, v.21 [65], using the generalized gradient approximation (GGA) [66], and include the spin-orbit coupling using the four component formalism and the standard basis set implemented in FPLO. Brillouin zone integrations were performed with a tetrahedron method using a mesh with $36 \times 36 \times 10$ subdivisions. The density of states (DoS) was computed starting with a fully converged electronic density and using a mesh with $96 \times 96 \times 26$ subdivisions in a single DFT step. Exemplary input files for calculations of models A and B can be found in the repository [67].

Figure 3(a) shows the band structure and DoS for the structural models $A$ and $B$. Both yield an insulating ground state with time-reversal polarization invariants $\mathcal{Z}_2 = [1; 110]$, and thus place $ZrTe_5$ in an STI phase. We have also computed the mirror Chern numbers $\mathcal{C}_x$ and $\mathcal{C}_z$ associated with the reflection symmetries $M_x$ and $M_z$, respectively (Fig. 2). These take the values $\mathcal{C}_x = 1$ and $\mathcal{C}_z = -1$, with definitions as in Ref. [68]. The main difference between models $A$ and $B$ lies in the number of low-energy electron pockets: while structure $A$ exhibits only one pocket, structure $B$ has several pockets. Yet, even the band structure of structure $A$ deviates from a plain vanilla massive Dirac model since our *ab-initio* calculations predict an indirect gap.

It has been argued that uniaxial strain applied along the $a$-axis can tune the system through the topological phase transition [22]. However, neither the band structure associated with structure $A$ nor $B$ is of a simple gapped Dirac type, since the gap is either indirect, or there are additional pockets. To thoroughly analyze the impact of strain on both structures, we supplement our model with a deformation $\delta_a = a - a_0$ along the $a$-axis. The strain-induced changes of $b$ and $c$ are taken into account via the Poisson coefficients $\gamma_{ab} = \gamma_{ac} = -0.25$,

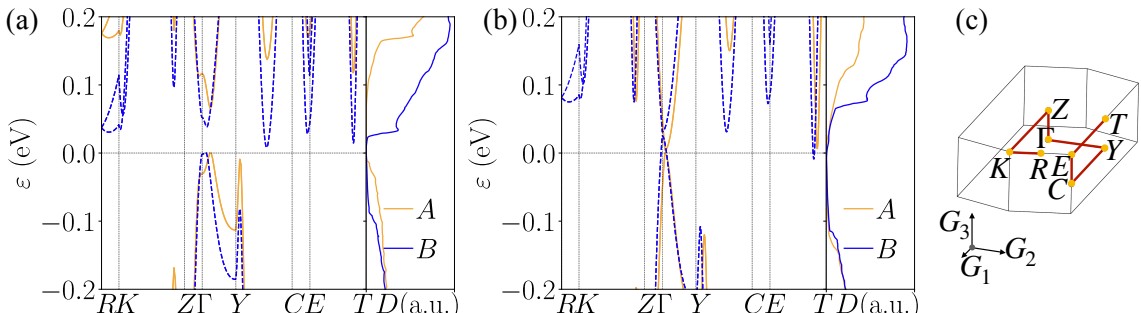

Figure 3: (a) Band structure and density of states (DoS) for structural models $A$ and $B$. (b) Bands and DoS close to the topological phase transition induced by uniaxial strain. The apparent different size of the electron and hole Fermi surface pockets visible close to the transition in model $B$ is compensated by the larger number of electron pockets and their larger elongation along the direction $G_2$. (c) Brillouin zone.

extracted from the theoretical results in Ref. [22]. We find that for both $A$ and $B$ structures, strain closes and reopens the gap at $\Gamma$, thereby changing the invariants to $\mathcal{Z}_2 = [0; 110]$ and $\mathcal{C}_x = \mathcal{C}_z = 0$. The critical strains are $\delta_a^{\text{cr}} = 1.7\%$ for $A$ and $0.6\%$ for $B$. This difference can be attributed to the different values of $a_0$.

In both cases, an electron pocket along the $E_0 - T$ line of the Brillouin zone lowers its energy towards the Fermi level. This is more pronounced in model $B$, for which the bottom of this additional band is already below zero energy at the critical strain $\delta_a^{\text{cr}}$. The band structure then is that of a semimetal with compensated electron and hole pockets. This is shown in Fig. 3(b), which compares the band structures at critical strain.

# 4 Microscopic origin of band structure variations

The discrepancy in how close the two structural models $A$ and $B$ are to a pure Dirac regime can be traced back to the crystal structure. More precisely, we find that the Dirac purity mainly depends on the different spacing between the $ZrTe_8$ polyhedra along the $c$-direction in the two models, i.e., on the Van der Waals gap $r_c$ (see also Fig. 2a).

The lattice parameters in structure $A$ are approximately one percent smaller than in structure $B$, but we have checked that the change of volume by itself does not change the band structures qualitatively (Appendix A). Also, the Van der Waals gap along the $b$-direction ($r_b$) is rather consistent between both structures and only differs by $\sim 0.2\%$. In stark contrast, the Van der Waals gap along the $c$-direction ($r_c$) is 4% smaller in $A$ than in $B$, which is the largest relative difference between the two structures.

To more systematically test the impact of $r_c$, we studied two auxiliary structural models. The model dubbed $A'$ has all structural parameters as in $A$, except for those defining $r_c$, which are taken as in model $B$. These parameters are the lattice parameter $c$ and the internal coordinate $z$ of the $Te_3$ atoms. The second auxiliary model, $B'$, is constructed in the opposite way: $B'$ has all parameters as in $B$ except for $c$ and $z$ of $Te_3$, which are taken as in $A$. Figure 4(a) shows the density of states $D(\varepsilon)$ for models $A$, $B$, $A'$ and $B'$. We find that is it mostly $r_c$ that determines whether or not $D(\varepsilon)$ increases rapidly above the gap, which in turn indicates the presence of additional electron pockets. This is further illustrated in Fig. 4(b), which shows the density of states at the Fermi energy as a function of $r_c$ for an electron carrier density of $1 \times 10^{19}$ electrons/cm$^3$.

The dependence of the electron pocket position on the Van der Waals gap can also be confirmed analyzing the effective hoppings between the Te atoms across the Van der Waals gaps. With this aim, we obtained tight-binding models by constructing Wannier functions associated with Te-5$p$ and Zn-4$d$ orbitals using the projective scheme implemented in FPLO [69]. In these tight-binding models, we replace the hoppings between the nearest-neighbor Te atoms across the Van der Waals gaps, i.e., we use these Te-Te hoppings from model $A$ in the tight-binding model $B$, and vice versa. We find that this replacement alone produces an energy shift of the electron pockets, which is even slightly larger than that shown in Fig. 3(a). This is detailed in Appendix B and it further confirms the Van der Waals gap $r_c$ as the key player bringing the low-energy band structure into, or out of, a pure Dirac regime.

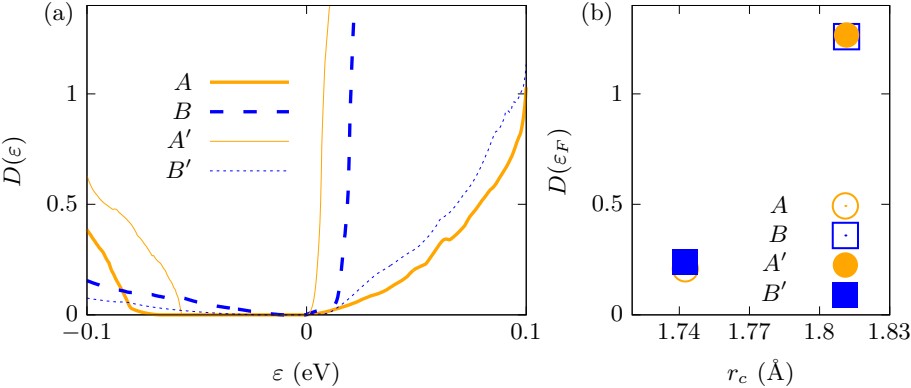

Figure 4: Panel (a): Density of states as a function of energy for the four models, $A$, $B$, $A'$, and $B'$. Panel (b): Density of states at the Fermi energy as a function of the Van der Waals gap $r_c$, for a doping of $1 \times 10^{19}$ electrons/cm$^3$ in all models. Models $A$ and $B$ correspond to our own crystal refinement at 50 K and to Ref. [64], respectively. Model $A'$ is obtained from model $A$ by using the lattice parameter $c$ and the $z$ coordinate of Te$_3$ atoms as in model $B$. Analogously, model $B'$ is obtained from model $B$ by using the lattice parameter $c$ and the $z$ coordinate of Te$_3$ atoms as in model $A$.

# 5 Dirac semimetal map

Having identified the Van der Waals gap $r_c$ as the most important parameter determining how close the low-energy band structure is to a pure Dirac regime, we now systematically map the phase space of the Van der Waals gap $r_c$ and uniaxial strain $\delta_a$ for the structural model $A$ describing our own samples. While a Dirac-like gap closing is guaranteed at the topological phase transition, we here aim at identifying an extended parameter regime without unwanted additional pockets close-by in energy, and in which there is a direct gap close to the $\Gamma$-point in reciprocal space.

To that end, we study two key quantities: the fundamental band gap $\Delta$, i.e., the gap in the DoS, and the difference between the gap at $\Gamma$ (denoted as $\Delta_\Gamma$) and $\Delta$. Figure 5(a) shows the fundamental band gap in the phase space of the Van der Waals gap $r_c$ and uniaxial strain $\delta_a$. At small Van der Waals gaps $r_c$, the gap closes and reopens as a function of applied strain $\delta_a$. This indicates that in this parameter regime there are no unwanted additional pockets within the energy gap present at $\Gamma$. The transition is therefore solely described by a pure Dirac gap closing. At large $r_c$, however, the system remains gapless after the gap closing, which in turn indicates the presence of additional electron and hole pockets.

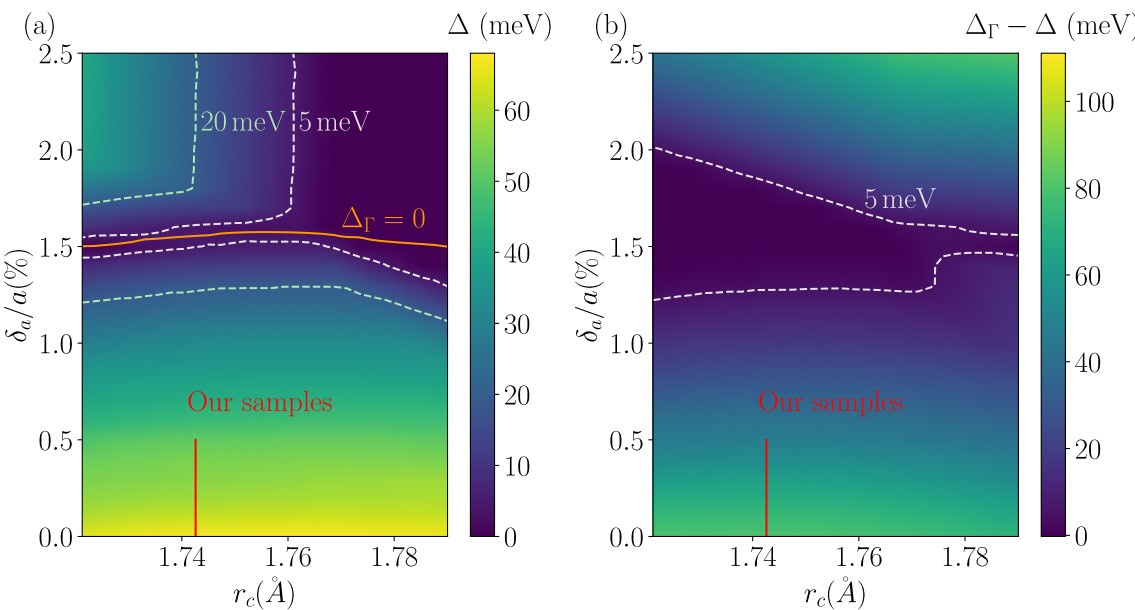

Figure 5: Left: Fundamental band gap in the phase space of the Van der Waals gap $r_c$ and uniaxial strain ($\delta_\alpha$). Dashed curves are isocontours of $\Delta$ while the continuous curve indicates $\Delta_\Gamma = 0$. Right: Difference between the fundamental gap and the gap at $\Gamma$. Dashed lines correspond to isocontours.

Figure 5(b) shows $\Delta_\Gamma - \Delta$, which serves as an indicator of the energy dispersion near $\Gamma$. This difference should vanish in an ideal (massless or massive) Dirac Hamiltonian. We observe that this only happens for small $r_c$, and in the presence of moderate uniaxial strain. At larger Van der Waals gaps $r_c$, a finite $\Delta_\Gamma - \Delta$ instead indicates an indirect gap, and therefore deviations from a pure Dirac regime. Depending on the parameter regime, this indirect gap can either result from additional pockets, or from the pocket close to $\Gamma$ being strongly deformed and having its minimum not exactly at $\Gamma$, see Fig. 3.

Overall, our calculations prove that ZrTe$_5$ can indeed be tuned to a pure Dirac semimetal regime by strain, albeit only for relatively small Van der Waals gaps $r_c$. At larger $r_c$, Dirac physics is instead masked by additional bands away from the $\Gamma$-point, but also by non-Dirac terms in the Hamiltonian deforming the pocket close to $\Gamma$. Still, our analysis shows that our own samples are located very well within the parameter range that allows to reach a pure Dirac regime by strain (see Fig. 3). This means that strain-tuning ZrTe$_5$ towards pure Dirac physics is not only possible, but also experimentally realistic.

## 6 Conclusion

We have analyzed the band structure of ZrTe$_5$ as a function of the microscopic atomic positions, showing that this material either exhibits a pure Dirac regime in which the low-energy band structure is solely described by a (weakly gapped) pure Dirac Hamiltonian, or a mixed regime in which additional trivial bands generally exist. This conclusion is obtained by combining original synchrotron measurements, extensive *ab-initio* calculations using DFT, and a comparison to existing literature.

Our results have important implications for future experiments on ZrTe$_5$ and other low-density semimetals [70]. If present, additional bands will typically mask the physics associated

with relativistic nodal quasiparticles. However, not all is lost: we find that external strain can serve as a powerful in-situ tuning knob for bringing the band structure into a pure Dirac regime. For ZrTe$_5$, strain along the $a$ axis can push the additional pockets far away from the Fermi level (i.e. to energies outside the window that is probed in the respective experiments). Our analysis furthermore uncovers an extended regime in the phase space spanned by the Van der Waals gap along the $c$-direction and strain along the $a$-direction in which pure Dirac physics is realized, but also shows that too large a Van der Waals gap can prevent a pure Dirac regime. This identifies the Van der Waals gap along $c$ as the most important factor impacting the Dirac character of the low-energy band structure in ZrTe$_5$.

Our study leaves several open questions for the future theoretical modelling of ZrTe$_5$. We have here limited ourselves to understanding how different structural parameters, within a range of parameters close to the experimentally-determined crystal structures, affect the low-energy electronic structure of ZrTe$_5$. This is a different question than converging towards a DFT prediction for the ground state of this compound, also including a better treatment of electronic correlation effects. One point to bear in mind is that all our DFT results are based on the GGA approximation for the exchange and correlation functional. In view of its limitations in the prediction of band gaps and the several suggestions in the literature that most available samples present properties more consistent with the WTI phase (rather than the STI predicted by GGA for the experimental crystal structure), it could be of interest to study the electronic structure of this compound with functionals known to describe more accurately the fundamental gap in weakly correlated semiconductors [71–73].

Future studies should also analyze how the charge carrier density and Fermi level are impacted by the interplay of crystallographic details and temperature. We have shown that the position in energy of trivial pockets is largely affected by the van der Waals gap $r_c$. The origin of the experimental dispersion in the values of $r_c$ is yet unclear. The two sets of experimental data used in this work rely on samples grown with different methods. Thus, performing a controlled analysis of growth conditions vs $r_c$ may be a valuable idea. It may also be fruitful to theoretically address the impact of defects on $r_c$ [74], as well as to consider possible structural variants of ZrTe$_5$ e.g., intercalation of alkali metas, where $r_c$ may potentially be significantly changed. Last, the experimental observation that the prominent Lifshitz transition at intermediate temperatures occurs at $90\,\mathrm{K}$ in structure A [24] while at $150\,\mathrm{K}$ in structure B [64] supports the idea of a strong interplay of crystallographic details and fermiology in ZrTe$_5$, and by extension possibly also in other Van der Waals materials.

In summary, our analysis establishes ZrTe$_5$ as a versatile platform in which strain can be used for the in-situ tuning of band structures, in particular enabling regimes with a pure Dirac band structure.

# Acknowledgements

**Funding information**   We acknowledge DESY (Hamburg, Germany), a member of the Helmholtz Association HGF, for the provision of experimental facilities. Parts of this research were carried out at PETRA III and we would like to thank P. Glaevecke for assistance in using beamline P21.1. Beamtime was allocated for proposal 11010761. We thank Ulrike Nitzsche for technical assistance. J.I.F. would like to thank the support from the Alexander von Humboldt Foundation during the part of his contribution to this work done in Germany and ANPCyT grants PICT 2018/01509 and PICT 2019/00371. E.N. and M.M. are fully funded by the Swedish Foundation for Strategic Research (SSF) within the Swedish national graduate school in neutron scattering (SwedNess) as well as the Swedish Research Council (VR) through a neutron project grant (Dnr. 2021-06157). G.G. and Q.L. were supported by the

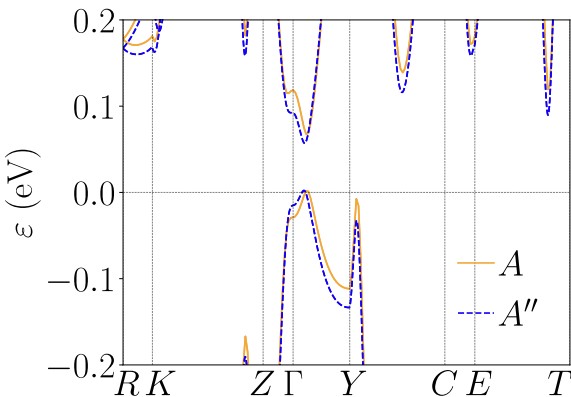

Figure 6: Band structures obtained for models $A$ and $A''$. The latter correspond to a structural with internal coordinates as model $A$ but lattice parameters $a$, $b$ and $c$ as in model $B$.

U.S. Department of Energy (DOE) the Office of Basic Energy Sciences, Materials Sciences, and Engineering Division under Contract No. DE-SC0012704. F.M. and T.M. acknowledge support from the Deutsche Forschungsgemeinschaft (DFG) through the Emmy Noether Programme ME4844/1-1. We also thank the Collaborative Research Center SFB 1143 (Project No. A04 and A05), and the Würzburg-Dresden Cluster of Excellence on Complexity and Topology in Quantum Matter – $ct.qmat$ (EXC 2147, Project No. 390858490).

# A Dependence of the low-energy band structure on the unit cell volume

In Fig. 6, we show the band structures of model $A$ and of a model $A''$ which has all internal coordinates as in $A$ but the lattice parameters $a$, $b$ and $c$ as in $B$. Thus, the volume of the unit cell of $A''$ is equal to that of $B$. Nevertheless, the low-energy bands are rather insensitive to this change. The Van der Waals gap of model $A''$ is $r_c \sim 1.763$Å, which is closer to that of model $A$ than of $B$ (these are given in Table 1). This is consistent with $r_c$ being the primary variable that controls the energies of the trivial electron pockets.

# B Dependence of the low-energy band structure on the Van der Waals gap $r_c$

In Fig. 7, we show how the position of the additional Fermi pockets is influenced by the effective coupling between the Te atoms across the Van der Waals gap $r_c$. The solid lines denote the bands of the original two models, labeled $A$ and $B$, on a section of the path between $R$ and $K$ in the BZ [compare with Fig. 3(a)]. They have been obtained by diagonalizing the Wannier Hamiltonians corresponding to these two models, using the kwant code [75]. To obtain the other two band structures, shown using dashed lines, we have swapped the hopping matrices connecting the neighboring Te atoms across the Van der Waals gap. Thus, model $A'$ denotes a Wannier Hamiltonian that is identical to that of $A$ in all respects, except that it contains the $Te_3$-$Te_3$ hoppings taken from model $B$. Similarly, model $B'$ contains the hoppings taken from model $A$. The relative shift in the energy of the additional Fermi pocket shows that the coupling

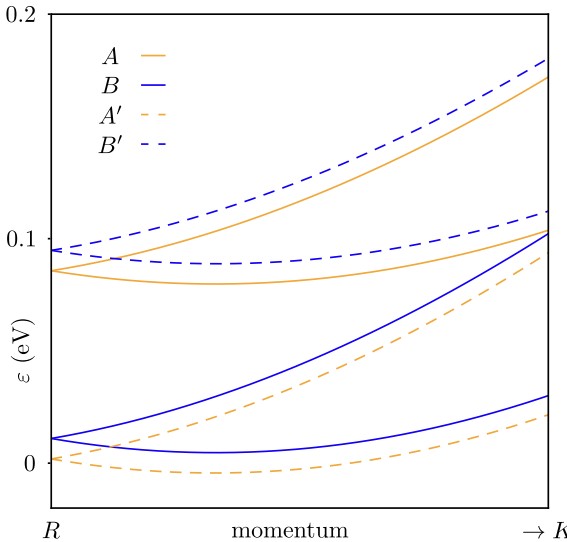

Figure 7: Band structures obtained from the Wannier Hamiltonians corresponding to models $A$ and $B$, as well as obtained from the modified models, $A'$ and $B'$.

between those Te atoms separated by the Van der Waals gap is principally responsible for how close the system is to a pure Dirac regime.

## C   Crystal growth

In this study, we have used a high-quality $ZrTe_5$ single crystal grown with high-purity elements (99.9 % zirconium and 99.9999 % tellurium) using the tellurium flux method. Further characteristics of the crystals have been reported in Ref. [24].

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
