# Peer review of "Engineering a pure Dirac regime in ZrTe$_5$"

_SciPost Physics, doi:SciPost Phys. 14, 066 (2023)_

## Round 1 · Referee Report · Anonymous (Referee 1) · 2022-7-20

Report

In the manuscript entitled « Engineering a pure Dirac regime in ZrTe5 » J. I. Facio and coworkers report an ab initio investigation of the band structure of ZrTe5. First, the authors have carefully determined the crystal structure and atomic positions by means of XRD. Second, the authors computed the band structure corresponding to different atomic positions and under strain. The core of the work is to isolate the parameters that allow not only the Dirac phase (closure of the direct gap at Γ) but also the absence of additional, and trivial, pocket at the Fermi level. Hence, this work deals with an important topic, the realization of a real Dirac semimetal phase where new exotic transport properties and quantum effects might be under the reach of experiments. I have found the manuscript well organized and written, and the conclusions sound solid.
I have nonetheless few questions that I kindly ask the authors to consider.

1) The authors propose the van der Waals gap rc to be “the most” important factor (maybe one important factor would be proper) determining the Dirac nature. In order to modify it, the authors play with the hopping integral between the Te atoms across the van der Waals gap. However, this parameter seems difficult to be controlled from an experimental point of view. Intercalation of alkali metal is, in my opinion a viable route, compatible at least with surface sensitive technique such as STS and ARPES. Can the authors propose alternative methods to experimentally confirm their prediction?
2) As the authors point out, ZrTe5 crystals of structure A and B display different temperature of the resistivity anomaly. Besides the structural differences, those crystals are synthetized via different methods, (Te flux, CVT) and they exhibit different defects. The importance of defects has been experimentally explored by B. Salzmann et al in Phys. Rev. Mater. 4 114201 (2020). Have the authors considered the role of defects, directly in the calculations and their influence in van der Waals gap rc ?
3) In the conclusions the authors leave the problem of electronic correlations, and the influence on the band gap, unsolved. Can the authors indicate the best strategies they would follow to tackle this problem?
4) In my opinion a few important additional references are missing. The evolution of the electronic properties of ZrTe5 with uniaxial strain has been experimentally investigated by P. Zhang et al., Nat. Commun. 12 406 (2021). The temperature evolution of the band structure and of the chemical potential have been first reported by G. Manzoni et al, Phys. Rev. Lett. 115 207402 (2015)

A) Minor: in the introduction the authors say “…powerful tool for in-situ tuning of the band structure such that all nontrivial pockets are pushed away from the Fermi energy…”. I guess the aim is to push the trivial pocket far from the Fermi energy, thus leaving only the topological one to be responsible for the conduction.

---

## Round 1 · Referee Report · Anonymous (Referee 2) · 2022-7-22

Strengths

- The paper is well structured, easy to follow the results and their discussion.
- Timely subject matter, discussing the tuning of topological properties of ZrTe5 by externally applied mechanical deformation.

Weaknesses

- The DFT code the authors use is not state of the art, as the authors themselves point out in the conclusions

Report

ZrTe5 is well known to be at the boundary of a topological phase transition, where weak and strong topological insulator and Dirac semimetal phases are relatively easily achieved by tuning the lattice parameters. The paper proposes a method to strain engineer the single particle band structure of ZrTe5, in order to realize a 3D Dirac semimetal phase. This sensitivity to relatively small changes in lattice parameters makes this material interesting, since it promises the realization of topological phase transitions under strains available in laboratory conditions. This is a very timely topic, as such it qualifies for inclusion in SciPost Physics.

Requested changes

- It would be worthwhile to discuss the results in light of other 3D Dirac semimetals, such as Na3Bi and Cd3As2, see for example:
Collins, J. L. et al. Electric-field-tuned topological phase transition in ultrathin Na3Bi. Nature 564, 390–394 (2018)
Jeon, S. et al. Landau quantization and quasiparticle interference in the three-dimensional Dirac semimetal Cd3As2. Nat. Mater. 13, 851–856 (2014)
Is the Dirac band also masked by other trivial bands in these cases? It seems to me that they are not. It would be to the benefit of the paper, if the authors expanded with some concrete examples on their statement in the introduction: “Most of the time, the Dirac points of 3D crystals occur at energies away from the Fermi level, or they are surrounded by additional trivial electronic pockets which obscure the unique features of linearly dispersing electrons”. Furthermore, in the case of thin Na3Bi, tuning of the topological phase by outside parameters (electric field in the case of the Jeon etal. results) is also possible, making it relevant to the current work. It might be fruitful to just mention other methods of tuning topological phases, outside of mechanical strain.
- Why is there a difference in the van der Waals gap in the c-direction in the two samples? What could be the reason for this? Maybe disorder, changes in stoichiometry?
- The assumption that the Poisson ratio in the ab and ac directions is the same seems dubious to me, since the crystal is clearly anisotropic in these directions. Why is this a valid assumption? How robust are the conclusions of the paper based on this assumption? For example, how much would a 1% and a 10% difference in Poisson ratio influence the topological phase transition?

---

## Round 1 · Referee Report · Anonymous (Referee 3) · 2022-8-19

Strengths

The authors propose a method (strain) to obtain a pure Dirac phase.
The measurements and calculations are sound, the material studied is interesting, and the search for pure Dirac phases is indeed timely. The combination of synchrotron diffraction experiments and ab-initio calculation is a strong point of the work, although the ab-initio calculations are only performed at the DFT level.

Weaknesses

My main criticism is about the robustness of the pure Dirac phase found with this method (see report).

Report

If the goal is to find a Dirac semimetal (i.e. Delta_Gamma = 0) without unwanted additional pockets, I think they should be looking for something more robust than just the Dirac-like gap closing of the phase transition. Otherwise, finding a sample that actually presents such a Dirac point would be just a matter of luck of having the sample at exactly the right lattice parameters. A tiny difference in the parameters would break the Dirac crossing, getting the sample out of the Dirac phase.

In other words, where is Delta_Gamma = 0 in the phase space of Fig. 5? This must be only a point (or a discrete line? surprisingly, they don't show Delta_Gamma in Fig. 5) making it almost impossible to tune the sample to land exactly on that particular interesting point. And if this is achieved, any tiny distortion will move the sample outside that spot. This is not a problem of the material, but rather of the method proposed in the paper. Any search for Dirac physics based on the Dirac gap closing in a topological phase transition is not practical for real applications.

What would be really interesting is to find a way to remove the trivial pockets without opening a gap at the Dirac point. Unfortunately, the method proposed here does both. Instead one needs to look for a Dirac crossing that is robust and protected by some symmetry. In other words, there should be a finite region (not discrete) in the phase space of Fig. 5 where Delta_Gamma is zero.

I am sure the paper deserves publication in some form, but I am unsure about the impact of the work for publication in SciPost Physics. Since this robustness is important for applications and this is key for the impact of the work, I think the authors should comment on the robustness of this phase and why it is interesting nevertheless, before it can be published.

I have a couple of other comments:

  • The authors do comment on the limitations of GGA for the prediction of the band structure. It would be good to do something beyond to see the extend of the many-body effects in this compound. It should not be a huge effort to calculate the band structure (of the unstrained material) with a method beyond GGA (a hybrid functional, for example).

  • Other points that lower the impact of the work are that 1) the topological transition upon strain is already known (as they correctly cite) and the only thing added here is the effect on the trivial pockets. And 2) the fact that strain can move pockets up and down in energy is in itself not very novel.

  • Figure 3: In the orange curve there seems to be a hole pocket but no electron pocket that compensates for it. In the blue curve, the whole pocket seems much larger than the electron pocket. Is the electron pocket that compensates the hole pocket somewhere else (not shown) within the Brillouin zone or is this a problem of convergence of the Fermi energy?

  • About this sentence: "While a Dirac-like gap closing is guaranteed at the topological phase transition, we here aim at identifying an extended parameter regime without unwanted additional pockets close-by in energy, and in which there is a direct gap close to the Gamma-point in reciprocal space." Why do they want to have a "direct gap close to the Gamma-point" and what do they mean with that?

  • Other computational parameters of the DFT calculations should be provided (not just the k meshes).

  • It seems like they use the term "pockets" for all the minima of the conduction band, even when they are relatively far from the Fermi energy. I wonder if this notation is appropriate. This minima would not appear as pockets in a plot of the Fermi surface, so I am not sure if one should call them "pockets". For example, Figure 3(a) is a semiconductor, not a semimetal, so speaking about "pockets" seems odd.

---

## Round 2 · Referee Report · Anonymous (Referee 4) · 2022-11-1

Report

The authors have addressed my questions and issues I raised. I recommend publication.

---

## Round 2 · Referee Report · Anonymous (Referee 5) · 2022-11-10

Report

I appreciate the efforts of the authors to reply to all my previous comments. I think the new formulations are more precise, consistent and explanatory.
But before publication, I would like to insist on the reproducibility issue. There must be other convergence parameters/cutoff values necessary to reproduce the results. For example parameters used to converge the basis set used by FPLO. In contrast to what the authors claim, the information they provide is typically not enough to reproduce DFT results.
After this information has been added, I can recommend publication in SciPost Physics.

---

## Round 2 · Referee Report · Anonymous (Referee 6) · 2022-11-22

Report

I thank the Authors for their reply that addresses my concerns and requests.
I think that the manuscript is suitable now for publication.

---

## Round 2 · Author Response

Dear editor,

we hereby resubmit our manuscript entitled “Engineering a pure Dirac regime in ZrTe5” to SciPost Physics. In doing so, we would like to thank all three referees for taking the time to carefully read our manuscript, and for raising a number of points in which the manuscript could be improved. We are delighted that all three referees found our work to deserve publication, and that is was praised as timely, well organized, and with solid conclusions. One referee even recommended publication in SciPost Physics already in their report. Below, we provide point-by-point responses to each point raised by the referees. In light of the referees’ remarks, we have modified parts of the manuscript, which we think brings out the main results even more clearly. We therefore believe that the manuscript is now appropriate for publication in SciPost Physics.

With best regards, the authors

Anonymous Report 1 on 2022-7-20

"In the manuscript entitled “Engineering a pure Dirac regime in ZrTe5” J. I. Facio and coworkers report an ab initio investigation of the band structure of ZrTe5. First, the authors have carefully determined the crystal structure and atomic positions by means of XRD. Second, the authors computed the band structure corresponding to different atomic positions and under strain. The core of the work is to isolate the parameters that allow not only the Dirac phase (closure of the direct gap at Γ) but also the absence of additional, and trivial, pocket at the Fermi level. Hence, this work deals with an important topic, the realization of a real Dirac semimetal phase where new exotic transport properties and quantum effects might be under the reach of experiments. I have found the manuscript well organized and written, and the conclusions sound solid."

-> We thank the Referee for taking time to review our work and for the useful comments.

"I have nonetheless few questions that I kindly ask the authors to consider. 1) The authors propose the van der Waals gap rc to be “the most” important factor (maybe one important factor would be proper) determining the Dirac nature. In order to modify it, the authors play with the hopping integral between the Te atoms across the van der Waals gap. However, this parameter seems difficult to be controlled from an experimental point of view. Intercalation of alkali metal is, in my opinion a viable route, compatible at least with surface sensitive technique such as STS and ARPES. Can the authors propose alternative methods to experimentally confirm their prediction?"

-> We thank the Referee for the remark. Intercalation of alkali metals is indeed a natural way to change the parameter rc. In addition, judging from the two sets of experimental data quoted in our work, assuming that both are correct, it seems that different samples may naturally present a distribution of values of rc. Regardless of whether that is so, performing a controlled analysis of different preparation method/conditions vs rc would also be a valuable idea. We now mention these ideas in a discussion added in the conclusion section.

"2) As the authors point out, ZrTe5 crystals of structure A and B display different temperature of the resistivity anomaly. Besides the structural differences, those crystals are synthetized via different methods, (Te flux, CVT) and they exhibit different defects. The importance of defects has been experimentally explored by B. Salzmann et al in Phys. Rev. Mater. 4 114201 (2020). Have the authors considered the role of defects, directly in the calculations and their influence in van der Waals gap rc ?"

-> We haven’t explored the role of defects but it is an interesting topic for future research. We now mention this, and cite the paper referred to by the referee, in a discussion added in the conclusion section.

"3) In the conclusions the authors leave the problem of electronic correlations, and the influence on the band gap, unsolved. Can the authors indicate the best strategies they would follow to tackle this problem?"

-> To our knowledge, there are least two ways in which the calculations could be improved. On the one hand, van der Waals correlations (which are a type of dynamical correlation effect) are likely important for determining the crystal ground state. We notice that in spite of this, for a given crystal structure, the band structure we obtain is essentially the same as found in Sci. Adv. 5(8), eaav9771 (2019) where van der Waals corrections were included. A second issue is the famous gap problem of local or semilocal functions like GGA. Here the best solutions found so far are exact-exchange methods (e.g., Phys. Rev. B 59, 10031, 1999), hybrid functionals (e.g., . Phys. Chem. Lett. 7, 20, 4165–4170, 2016), or GW approaches (see Ref. PRB 84, 041109(R), 2011). The latter reference, for example, illustrates different cases in which the topological properties are wrongly predicted by semilocal functions. It would be certainly of interest to study the electronic structure of ZrTe5 with these methods which often substantially increase the computation time and deserve a dedicated study.

In our opinion, it is sufficient for this paper to raise the point that this is a material where LDA/GGA may not be enough to precisely charter the detailed boundary of the topological phase transition, without entering into details of how this could be improved. However, following the Referee question, we have added citations to the papers in the paragraph above, in case they are of use for the interested reader.

"4) In my opinion a few important additional references are missing. The evolution of the electronic properties of ZrTe5 with uniaxial strain has been experimentally investigated by P. Zhang et al., Nat. Commun. 12 406 (2021). The temperature evolution of the band structure and of the chemical potential have been first reported by G. Manzoni et al, Phys. Rev. Lett. 115 207402 (2015)"

-> We thank the Referee for pointing out these relevant references, which we have now incorporated in the intro duction of the paper.

"A) Minor: in the introduction the authors say “...powerful tool for in-situ tuning of the band structure such that all nontrivial pockets are pushed away from the Fermi energy...”. I guess the aim is to push the trivial pocket far from the Fermi energy, thus leaving only the topological one to be responsible for the conduction."

-> We thank the Referee for the remark. The aim is indeed to push the trivial pockets far from the Fermi energy, where far means “further than all relevant energies involved in probing the system”. We have amended the relevant sentence, and added an explanatory half-sentence in the conclusions.

Anonymous Report 2 on 2022-7-22

"ZrTe5 is well known to be at the boundary of a topological phase transition, where weak and strong topological insulator and Dirac semimetal phases are relatively easily achieved by tuning the lattice parameters. The paper proposes a method to strain engineer the single particle band structure of ZrTe5, in order to realize a 3D Dirac semimetal phase. This sensitivity to relatively small changes in lattice parameters makes this material interesting, since it promises the realization of topological phase transitions under strains available in laboratory conditions. This is a very timely topic, as such it qualifies for inclusion in SciPost Physics."

We thank the Referee for the time to review our work and for its positive appreciation.

"REQUESTED CHANGES - It would be worthwhile to discuss the results in light of other 3D Dirac semimetals, such as Na3Bi and Cd3As2, see for example: Collins, J. L. et al. Electric-field-tuned topological phase transition in ultrathin Na3Bi. Nature 564, 390–394 (2018) Jeon, S. et al. Landau quantization and quasiparticle interference in the three-dimensional Dirac semimetal Cd3As2. Nat. Mater. 13, 851–856 (2014) Is the Dirac band also masked by other trivial bands in these cases? It seems to me that they are not. It would be to the benefit of the paper, if the authors expanded with some concrete examples on their statement in the introduction: “Most of the time, the Dirac points of 3D crystals occur at energies away from the Fermi level, or they are surrounded by additional trivial electronic pockets which obscure the unique features of linearly dispersing electrons”. Furthermore, in the case of thin Na3Bi, tuning of the topological phase by outside parameters (electric field in the case of the Jeon etal. results) is also possible, making it relevant to the current work. It might be fruitful to just mention other methods of tuning topological phases, outside of mechanical strain."

-> We thank the Referee for the remark. To best of our knowledge, indeed the band structure of Na3Bi and of Cd3As2 is not masked by other trivial bands. Based on this comment and others from the Report 3, we have improved our introduction connecting it with other systems where Dirac physics can be studied. We have accordingly added citations to the works mentioned by the Referee and others.

"- Why is there a difference in the van der Waals gap in the c-direction in the two samples? What could be the reason for this? Maybe disorder, changes in stoichiometry?"

-> The origin of the difference in rc between the samples is a very pertinent question for which we unfortunately do not have an answer. One reason could be that the growth methods of the two samples are different. A more conclusive answer would, however, require a detailed controlled analysis of preparation method/conditions vs rc. We have added a discussion in the conclusion section about the open question that represents the origin in the dispersion in the values of rc and possibilities to control it.

"- The assumption that the Poisson ratio in the ab and ac directions is the same seems dubious to me, since the crystal is clearly anisotropic in these directions. Why is this a valid assumption? How robust are the conclusions of the paper based on this assumption? For example, how much would a 1% and a 10% difference in Poisson ratio influence the topological phase transition?"

-> While in the reported results we have used this choice, a detailed analysis of the theoretical data shown in 10.1126/sci- adv.aav977, indicates that their theoretical values are γab = −0.2 and γac = −0.25. We have performed additional calculations with this second choice of parameters which yield essentially identical results. For instance, in the cases the topological phase transition occurs at the same value of the deformation, or the bands for a given value of deformation (δa) are rather insensitive to fix the lattice parameters b and c using same or different Poisson ratios.

Anonymous Report 3 on 2022-8-19

"If the goal is to find a Dirac semimetal (i.e. ∆Γ = 0) without unwanted additional pockets, I think they should be looking for something more robust than just the Dirac-like gap closing of the phase transition. Otherwise, finding a sample that actually presents such a Dirac point would be just a matter of luck of having the sample at exactly the right lattice parameters. A tiny difference in the parameters would break the Dirac crossing, getting the sample out of the Dirac phase. In other words, where is ∆Γ = 0 in the phase space of Fig. 5? This must be only a point (or a discrete line? surprisingly, they don’t show ∆Γ in Fig. 5) making it almost impossible to tune the sample to land exactly on that particular interesting point. And if this is achieved, any tiny distortion will move the sample outside that spot. This is not a problem of the material, but rather of the method proposed in the paper. Any search for Dirac physics based on the Dirac gap closing in a topological phase transition is not practical for real applications. What would be really interesting is to find a way to remove the trivial pockets without opening a gap at the Dirac point. Unfortunately, the method proposed here does both. Instead one needs to look for a Dirac crossing that is robust and protected by some symmetry. In other words, there should be a finite region (not discrete) in the phase space of Fig. 5 where ∆Γ is zero. I am sure the paper deserves publication in some form, but I am unsure about the impact of the work for publication in SciPost Physics. Since this robustness is important for applications and this is key for the impact of the work, I think the authors should comment on the robustness of this phase and why it is interesting nevertheless, before it can be published."

-> We thank the Referee for carefully reviewing our work. The Referee is correct that being close to a topological phase transition may not be beneficial for realizing a gapless Dirac Hamiltonian. As the Referee points out, this is a region of measure zero in the phase diagram. It can thus, fundamentally speaking, never be reached in any experiment (i.e. gap closed and chemical potential exactly at the nodal crossing) - even if the crossing were symmetry-protected. This is, however, not the goal of our study. We are interested in a regime in which the physics is dominated by „Dirac-like“/relativistic effects, such as (to quote our introduction) „These electronic states have large velocities, long wavelengths, and are robust against backscattering“. This physics can also be reached if the chemical potential is in the linear band regime, and above (or below) a sufficiently small gap. In that sense, the presence of a sufficiently small gap is in no way detrimental to, e.g., the intriguing transport physics observed in experiments. In theory, one can for example tune the gap between a small finite value and zero, and see that many transport properties are largely unaffected (one important class of exceptions are zero-field anomalous responses, but magnetotransport is for example much less sensitive to whether or not there is a small gap). To illustrate this line of reasoning, we recall that it is theoretically certain that graphene has a gap. Yet, this gap is experimentally so small that it does not affect observable quantities. This prominent example shows that an exactly vanishing gap is not a prerequisite for the observation of Dirac physics. What does certainly harm such observations is if the Dirac or Dirac-like bands coexist with trivial bands that mask Dirac(-like) physics. In summary, we are not interested in providing a recipe to reaching a Dirac point exactly, but reaching a regime dominated by Dirac physics.

Following the Referee’s remark, we have modified the introduction so that the context of ZrTe5 as an interesting compound to study Dirac physics is more explicit. We have also modified Fig. 5 indicating the line where ∆Γ is zero.

"I have a couple of other comments: - The authors do comment on the limitations of GGA for the prediction of the band structure. It would be good to do something beyond to see the extend of the many-body effects in this compound. It should not be a huge effort to calculate the band structure (of the unstrained material) with a method beyond GGA (a hybrid functional, for example)."

-> We agree that it is certainly of interest to study this material beyond the GGA approximation. As we explain in our response to referee 1, point 3, this question is actually one with many facets. There exists for example a number of techniques that go beyond GGA, and that are believed to improve on different of the shortcomings of GGA. We therefore believe that while performing one such calculation might be regarded as “not a huge effort”, an honest comparative study of the compound with other exchange and correlation functionals is an issue that deserves a dedicated and separate work.

"- Other points that lower the impact of the work are that 1) the topological transition upon strain is already known (as they correctly cite) and the only thing added here is the effect on the trivial pockets. And 2) the fact that strain can move pockets up and down in energy is in itself not very novel."

-> We thank the Referee for their opinion. We would like to explain why we disagree. While the topological phase transition was previously studied, e.g. in Sci. Adv. 5(8), eaav9771 (2019), it has gone without notice that the actual gap for charge excitations can close before the transition happens due to the relative shift of an electron pocket away from Gamma. This is the case in particular for the crystal structure theoretically studied in that reference. From an experimental point of view, the fact that the transition may be masked by an insulator to semimetal transition is of crucial importance. In addition, our results show that, quite unexpectedly, the scenario induced by uniaxial strain sensitively depends on other crystal structure parameters than those more directly affected by the uniaxial strain: the van der Waals gap along the c-direction. It is tempting to imagine connections between our results and various ongoing puzzles about ZrTe5, such as the very different temperature reported for a Lifshitz transition, or the appearance and evolution of various Shubnikov de Hass oscilations under uniaxial strain (arXiv 2201.04049). We thus believe that our results are of interest for the broad community presently charting the phase diagram of ZrTe5 and, more generally, for researchers studying topological electronic phases in van der Waals compounds.

We have modified the introduction (in the second paragraph of the third page) in order to emphasize the novelty of our results.

"- Figure 3: In the orange curve there seems to be a hole pocket but no electron pocket that compensates for it. In the blue curve, the whole pocket seems much larger than the electron pocket. Is the electron pocket that compensates the hole pocket somewhere else (not shown) within the Brillouin zone or is this a problem of convergence of the Fermi energy?"

-> We thank the Referee for the observation. As for the orange curve, the lack of compensation that the Referee notices is due to Fermi energy convergence. We have improved the density of the k-mesh used to improve this aspect of the calculation. As for the blue curve, there are no additional pockets. While the hole and electron pockets look uncompensated in the present plot, this is not the case once the shape of these pockets along other directions and their number is taken in consideration. In particular, it happens that the pockets at the Brillouin zone border are more elongated along the direction G2 than the pocket at Gamma. We have added a remark in the caption of this Figure to prevent further confusion regarding this point.

"- About this sentence: “While a Dirac-like gap closing is guaranteed at the topological phase transition, we here aim at identifying an extended parameter regime without unwanted additional pockets close-by in energy, and in which there is a direct gap close to the Gamma-point in reciprocal space.” Why do they want to have a “direct gap close to the Gamma-point” and what do they mean with that?"

-> By direct gap at Gamma, we mean that the smallest energy gap between conducting and valence states occurs (exactly at) at Gamma. When massive, the ideal Dirac Hamiltonian presents a direct gap. Therefore, having an indirect gap is an undesired deviation from the ideal Dirac Hamiltonian.

"- Other computational parameters of the DFT calculations should be provided (not just the k meshes)."

-> In the first version of the draft, in addition to the k meshes, we provided methodological aspects such as the treatment of the spin-orbit coupling, how the density of states were computed, how Brillouin zone integrations were performed and the DFT code used. In our experience, this information (together with the atomic positions given in Table I) is typically enough to ensure the reproducibility of our results. However, if the Referee could be more specific about what additional information we should provide we would of course be happy to add it.

"- It seems like they use the term "pockets" for all the minima of the conduction band, even when they are relatively far from the Fermi energy. I wonder if this notation is appropriate. This minima would not appear as pockets in a plot of the Fermi surface, so I am not sure if one should call them "pockets". For example, Figure 3(a) is a semiconductor, not a semimetal, so speaking about "pockets" seems odd."

-> We thank the Referee for the remark. Since samples of ZrTe5 with varying amounts of doping can be experimentally studied, we would prefer to keep using the notation that the Referee correctly summarizes.

---

## Round 2 · List of Changes

- We extended the introduction (two paragraphs, starting with "The age of Dirac semimetals..." until "...in the linear band regime." to discuss when gaps at the nodal point become relevant, and to connected our study for ZrTe5 with other systems where Dirac physics can be studied.

- We extended the discussion on the novelty of our results with regard to strain, Van der Waals gaps and additional pockets (paragraph starting with "Recent work showed that strain..." and ending with "Van der Waals gaps in the crystal structure").

- We have modified the caption of Fig. 3 in order to comment on the size and shape of the Fermi pockets.

- We have modified Figure 5 to show the gap closing line.

- We have expanded the conclusions (paragraph starting with "Future studies should also..." and ending with "other Van der Waals materials."), in particular now including a discussion of growth conditions, defects, intercalation of alkali atoms, and further comment on the strong interplay of crystallographic details and terminology in ZrTe5.

---

## Round 3 · Author Response

Dear editor, dear referees,

we thank all referees for their careful work. We are delighted that two out of three referees have recommended publication in SciPost Physics as is. The third referee has a final request, namely that we provide additional details on our numerical implementations (especially convergence parameters/cutoff values). In response, we have

(i) specified in the main text that we use the standard basis set implemented in FPLO,
(ii) provided exemplary input files in an online repository https://doi.org/10.5281/zenodo.7395945, now cited as reference 67.

We are confident that this will provide all input needed for colleagues to reproduce our results, and hope the the paper can now be published in its present form. We would like to seize the opportunity to again thank all the referees for their valuable input, which we feel was very helpful in improving the paper.

On behalf of the authors,
Tobias Meng

---

## Round 3 · List of Changes

(i) specified in the main text that we use the standard basis set implemented in FPLO,
(ii) provided exemplary input files in an online repository https://doi.org/10.5281/zenodo.7395945, now cited as reference 67.

Both changes can be found on p. 4 in the first paragraph of Sec. 3.

---

## Editorial Decision

published